# Assessing the Hazard Degree of Wadi Malham Basin in Saudi Arabia and Its Impact on North Train Railway Infrastructure

Fatmah Nassir Alqreai [1] and Hamad Ahmed Altuwaijri [2,*]

1 Department of Geographic Information Systems, College of Social Sciences, University of Jeddah, Jeddah 23445, Saudi Arabia; fnalqreai@uj.edu.sa
2 Department of Geography, College of Humanities and Social Sciences, King Saud University, Riyadh 11451, Saudi Arabia
* Correspondence: haaltuwaijri@ksu.edu.sa

**Abstract:** The North Train Railway in the Kingdom of Saudi Arabia (KSA) extends over vast areas, crossing various terrains, including valleys, sand veins, plateaus, and hills. Therefore, the railway was designed and implemented to suit this environmental diversity under the highest safety standards. However, the railway may be subject to hazards for various reasons. In general, the possibility of direct surface runoff disasters increases if there are residential areas and facilities within the boundaries of drainage basins. Therefore, these areas should be studied, and the degree of hazard in drainage basins should be accurately determined. Hence, this study analyzed the degree of risk of 14 drainage basins affecting the North Train Railway within the Wadi Malham drainage basin. The risk degree model was used with eight parameters that have hydrological indications to give an idea of the behavior of direct surface runoff and alter the risk of direct surface runoff. We found that 28.57% of the total basins in the study area have overall score values indicating they are high-risk basins, namely basins 6, 7, 13, and 14. It is recommended to estimate the rainfall depth during different return periods, analyze soil permeability and land use classification in the study area, and apply hydrological modeling of drainage basins, which contributes to estimating the volume and peak of direct surface runoff in such arid and semi-arid environments that do not contain hydrometric stations to monitor the runoff.

**Keywords:** geographic information system; drainage basins; morphometric analysis; hazard degree; North Train Railway; Wadi Malham; ALOS PALSAR

## 1. Introduction

The railway network in the KSA consists of lines for transporting passengers, goods, petrochemical products, petroleum, and minerals. The North Train Railway is one of the pillars of this system, as it consists of a line for passengers that is 1250 km long, in addition to a 1550 km shipping line [1]. Any transportation will contribute to and impact the development of countries by connecting cities with each other [2]. The North Train Railway connects five regions, starting from Riyadh, the capital of Saudi Arabia, to the end of the north border of the county, which provides access to many cities and the countryside without using cars [1]. The North Train Railway provides access to higher education, employment, and trade exchange [3]. Here lies one of the challenges, as the North Train Railway extends over vast areas and crosses various terrains, including valleys, sand veins, plateaus, and hills. Therefore, the railway was designed and implemented to suit environmental diversity according to the highest safety standards. However, the railway may be subject to hazards for multiple reasons. It is likely to be subject to geomorphological and hydrological risks. This depends on the climate and terrain of the area where it is located [4], in addition to human errors that may occur when planning or implementing such projects. The railways in the Kingdom of Saudi Arabia were damaged in various

places due to torrential rains. In 2018, a part of the North Train Railway within the borders of the Wadi Bayer Basin in the Qurayyat Governorate was damaged. Also, in 2017, a part of the Riyadh–Dammam railway drifted and caused a train to derail, causing injury to passengers and damage to infrastructure due to direct surface runoff in Dhahran's southern (armored) valley [4].

Since the 1930s, people have been interested in studying drainage basins as geomorphological units. Then, people began using morphometric analysis for analyzing and quantifying drainage networks. Horton, a leader in this field, was the first to conduct this analysis. Afterward, the methods of morphometric analysis, which describe the drainage basins and networks, were developed through research, and attempts were provided by various researchers, such as Strahler, Shreve, Chorley, and others [5].

Strahler [6] defined morphometrics as "the science of measuring geometric properties of the Earth's surface resulting from the river erosion system" [7]. Morphometric measurements are some of the essential geomorphological applications. They provide a database for studying drainage basin properties and drainage network properties and simulating their hydrological behavior [8]. Moreover, these measurements are among the most critical factors affecting the degree of natural hazard, especially the hazard of direct surface runoff in drainage basins [9]. Through these measurements, the features of drainage basins can be understood and their degrees of hazard can be determined [10]. These parameters include the spatial, morphological, and topographic features of the drainage basin and drainage network [11].

It is known that drainage basins in arid and semi-arid environments do not flow throughout the whole year. There can be no direct surface runoff for long periods, as rain may fall but most of the water is lost in the soil. However, in other years, the basins may witness rain that is rare in its intensity where there is direct surface runoff, which may cause damage and destruction. In general, the possibility of direct surface runoff disasters increases if there are residential areas and facilities within the boundaries of the drainage basins. Therefore, these areas should be studied, and the degree of hazard in drainage basins should be accurately determined.

Many initiatives have emerged by the authorities and ministries to achieve the goals set by the Kingdom of Saudi Arabia within the framework of Vision 2030 for the Kingdom to be among the ranks of developed countries. Much of this framework revolves around ensuring safety and protection for individuals and property, as well as the continued development and sustainability of infrastructure and improvement in the quality of public facilities and the performance of services [12]. These aspects are considered within the direction and endeavor of this study. Thus, the importance of the study is represented in developing an accurate and scientific perception of the degree of danger of direct surface runoff in the drainage basins affecting the North Train Railway within the Wadi Mulham basin. This study can also be an applied scientific addition to environmental studies that contribute to authorities' decision making related to the planning, development, and protection of railways from the dangers of direct surface runoff. By reading the soil maps covering the study area, which the Ministry of Environment, Water, and Agriculture of Saudi Arabia (1984) produced, it became clear that the soil units within the area are loam soils. This is a type of hydrological soil group B that is characterized by having moderate to good drainage [13] and contributes to the generation of direct runoff. Based on this information and due to the strategic importance of the North Train Railway, it can be said that a limitation of existing studies is the lack of analysis of hazard areas around the North Train Railway in the basin of Wadi Malham. Thus, the study aimed to conduct morphometric analyses using a suitable digital elevation model to analyze drainage basins affecting the North Train Railway within Wadi Malham, classify the degree of hazard of direct surface runoff in basins, and map direct surface runoff hazards.

## 2. Classification of Direct Surface Runoff Hazard in Drainage Basins

The most significant direct surface runoff hazard is flooding, along with destruction of and damage to many natural and human features often caused by floods. Thus, floods are considered a disaster due to the effects they cause. Thus, it is essential to study and classify the drainage basins based on how hazardous the surface runoff is. Determining priorities helps to achieve effective management and decision making [14].

### 2.1. Classification Methods for Direct Surface Runoff Hazards in Drainage Basins

The literature review indicates that researchers have used various methods to evaluate the potential hazards of direct surface runoff. These methods are divided into qualitative, semi-quantitative, and quantitative [15] methods. For example, some researchers apply qualitative methods, such as classifying direct surface runoff hazards and defining them by using qualitative descriptions such as "very high", "high", "medium", "low", and "very low" or the like. These classifications are considered guiding models because they depend on the judgment and experience of the people who create them. Their details depend on historical records of direct surface runoff events in the region [16]. As mentioned by [15], adding weights and ranks to qualitative methods changes them to semi-qualitative methods, including the morphometric hazard degree, which is considered to be commonly used in studying the hazards of surface runoff [17], especially in areas where no hydrometric stations are available [4]. This is in addition to the methods mentioned by [18] such as multicriteria decision making (MCDM), including the analytical hierarchy process (AHP) [4,19] and the analytic network process (ANP), which is similar to AHP but deals with various consequences and interactions in the network [20]. Furthermore, there is also the method of arranging priorities through similarity with fuzzy TOPSIS [21].

Concerning quantitative methods, as mentioned by Chen et al., quantitative methods often depend on numerical modeling in analyzing the hazard of direct surface runoff [22], including, for example, hydrologic models [23] and hydraulic models [24,25]; there are statistical methods, which stand as quantitative methods in setting the plans for direct surface runoff susceptibility in addition to the evaluation of its hazard [26], including, for example, logistic regression (LR) [27]. Also, quantitative methods include machine learning, a branch of artificial intelligence, including artificial deep neural networks (DNNs) [17].

As is mentioned above, the semi-quantitative methods are used broadly, mainly when there are no data of high quality (measured through hydrometric stations), as is the case in the area under study. It has also been proven that these methods are helpful for regional and large-scale studies, as mentioned by the authors of [28–30] and documented in [15]. The method of hazard degree was selected as it is considered one of the most important scientific methods which contributes to classifying the direct surface runoff hazard in drainage basins, which can be used in drainage basins in arid and semi-arid environments [4].

### 2.2. Direct Surface Runoff Hazard Studies

Many previous studies that are directly related to the subject matter of this study concerning the classification of drainage basins in terms of direct surface runoff hazard degree were reviewed. This was carried out by applying the hazard degree model on standards and some morphometric measurements of drainage basins and networks. Various researchers have used this methodology in studying the direct surface runoff hazard, including [4,17,31,32].

The authors of [31] aimed to evaluate the hazard degree of floods to the main basin and sub-basins of Wadi Al-Leith. They determined the adequate procedures of mitigation and methods of using the water of surface runoff in these basins by adopting linear, areal, and topographical morphometric characteristics measured by ArcGIS and ASTER with a spatial resolution of 30 m. A total of twenty-five morphometric variables were calculated; then, nine morphometric variables were reached, including those morphometric variables with a direct relationship and inverse relationship with flood severity, to apply the model of hazard degree. The authors of this study presented their results and recommendations, the

most important of which was that by classifying the hazard categories into three groups, high, moderate, and low risk, there were five high-risk sub-basins, four low-risk sub-basins, and one moderate-risk sub-basin. One of the most important recommendations of this study was to build dams and barriers in high-risk basins that do not have an opportunity to feed the groundwater aquifer to benefit from its running water at the intersection points of the fourth- and fifth-order sewages. In the study of [17] on the evaluation of the hazard of floods and vulnerability of dry docks, the researchers calculated 14 morphometric variables. ArcGIS and ASTER were once again used for measurement with a spatial resolution of 30 m for sixteen fourth-order sub-basins in Wadi Rajil and five sub-basins, which are related to Wadi Al Waheeda in Jordan. A total of 11 morphometric variables were used, including those that have direct and inverse relationships with the severity of floods in order to apply the method of hazard degree. Moreover, three morphometric variables were used to apply the Al Chamy method. Using the two methods, the researchers could evaluate the severity of floods in the drainage basins. One of the most noticeable results obtained in the study was that plans were set which classified the drainage basins into different hazard degrees. Based on the method of hazard degree, the percentage of drainage basins that are highly and severely vulnerable to floods is estimated at 50% of the sub-basins in Wadi Rajil and 80% of the sub-basins in Wadi Al Waheeda. The study recommended that it is necessary to take protective measures in these basins to protect cities, roads, and infrastructure from floods, and preserve future development. Elsadek et al. [33] used the same methodology with a variation in terms of applying the model to three morphometric variables that have a direct relationship only with the severity of floods. They were calculated using the SRTM with a spatial resolution of 90 m. This was to produce a plan to identify the severity of floods in the 70 sub-basins of Wadi Qena in Egypt. Hazard was classified into five categories. They concluded that the prevailing feature of the hazard degree was the low and moderate risks. The percentage of moderate- to high-risk basins was 48.6% of the total number of sub-basins and the percentage of lower-risk sub-basins was 51.4%. In the study of [4], ArcGIS, WMS, and ALOS PALSAR were used with a spatial resolution of 12.5 m to apply the same model of hazard degree on 24 morphometric variables as per its direct or positive relationship with the severity of floods. The study was applied to five drainage basins in Al Qurayyat. Its direct surface runoff crosses the North Train Railway and some areas of the Governorate. The most significant results indicated very high risks in the drainage basins of Wadi Al Makhrouq and Bayer, high risks in Wadi Sarmada, moderate risks in Wadi Husaydah Al Gharbia, and very low risks in Wadi Umm Nakhila. The study recommended that long-term plans should be implemented to address the most dangerous sites and that the company that owns the North Train Railway project should consider the suitability of the site of the railway line from Riyadh in the south to Hasidah in the north. Moreover, in [34], ArcGIS and WMS were used to implement the methodology of evaluation and modeling of floods and their risks within three sub-basins in Wadi Al Azariq in Egypt. A total of 38 morphometric variables were calculated by using topographic maps and recent satellite images. Nine variables were approved, some of which have a direct relationship and some of which have negative relationships with the severity of floods, to implement the method of hazard degree to evaluate the hazard of these floods in the region. Moreover, hydrographic charts were created through hydrological modeling in different return periods in drainage basins to manage the region, use its water, and find out more details about the areas vulnerable to floods. By applying the hazard degree, the study concluded that the sub-basins of Wadi Al Azariq can be classified in terms of two degrees: low risk, to which sub-basin (1) is related, and high risk, to which sub-basins (2) and (3) are related. Therefore, sub-basin (1) has a fantastic opportunity to feed the layers of groundwater compared to sub-basins (2) and (3) which have a high possibility of surface runoff. However, flooded areas were determined and the most prominent recommendations of the study were to construct dams in order to protect the area from floods and to give an opportunity to re-feed the layers of groundwater. These studies were similar in terms of using the hazard degree model to measure the severity of

direct surface runoff in unmeasured drainage basins and classify them. They were only different in choosing the morphometric standards used in the measurement and spatial resolution of digital elevation models.

This study depended on previous studies, which applied scientific methodologies to assess the hazard of direct surface runoff in unmeasured drainage basins using the characteristics of drainage basins and networks. Previous studies tackled the method of assessing the hazard degree of direct surface runoff in drainage basins and the capability to compare these basins; this was based on the recommendations of researchers, which showed the importance of studying drainage basins in order to achieve public safety and contribute to proper planning and development of facilities located in drainage basins and demonstrated the constant progress of the development wheel. This study adopted the hazard degree model to classify the drainage basins and to determine which areas are more vulnerable to the risk of direct surface runoff. All of the above were applied within the boundaries of Wadi Malham, located in Riyadh, and affected the North Train Railway. The possibility of direct surface runoff disasters generally increases where residential areas and facilities are located within the boundaries of drainage basins, as this affects the hydrological behavior of surface runoff. Thus, this should be studied and its characteristics accurately determined.

## 3. Study Area

The study area (Figure 1) is geographically located between latitudes 24°55′35″ and 25°23′45″ N and longitudes 45°55′0″ and 46°33′0″ E in the center of the KSA in Riyadh. In particular, this area lies within the drainage basin of Wadi Malham, affecting the North Train Railway where the direct surface runoff passes through the main stream and some sub-streams of the Wadi Malham basin through the railway structure. Wadi Malham is one of the valleys of Jabal Tuwaiq in the eastern section of the plateau of Najd, where sedimentary rocks spread and it is calcareous, sandy, and loamy [34]. It is a sub-valley whose flow path is consistent with the inclination of rock layers in terms of their direction [35]. In describing this valley, Khamis mentioned in his book that it is a massive valley with many names, including Wadi Abu Qatada and Wadi Huraymila [36]. The valley descends from Jabal Tuwaiq in the eastern direction, meets many tributaries, and extends to the ports of Benban. Then, it continues its path until it reaches the mouth in southern Rawdat Al Khafs. The study area is located within the scope of a desert climate, where temperatures rise in summer, decrease in winter, and are moderate in spring and autumn [37]. This area is affected by the Mediterranean swale, which brings cyclonic rains in winter, early spring, and late autumn [34]. The average precipitation per year is 85 mm. The average temperature in Riyadh is 25 degrees Celsius. It rises to more than 45 degrees Celsius and decreases in winter to 0 degrees Celsius. The humidity level is almost 33% [38].

Drainage basins affecting the North Train Railway within the boundaries of Wadi Malham were studied. The area of these basins is 2.59 km$^2$ or more due to the unit hydrograph theory [39], which is suggested to be used in the hydrology of the study area. The number of basins based on the above is 14.

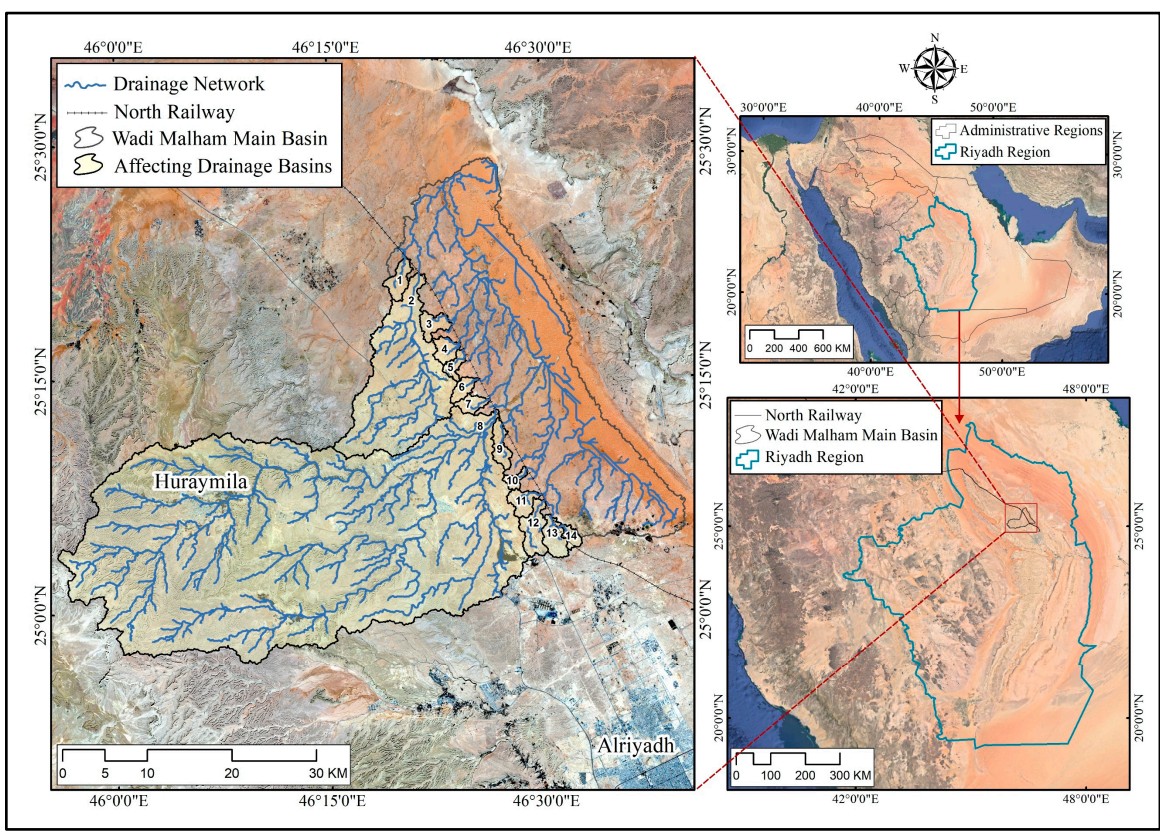

**Figure 1.** Study Area.

## 4. Preparation of Data

### 4.1. Types and Sources of Data

To carry out this study, various data were used from different sources as follows:

#### 4.1.1. Remote Sensing Data

A digital elevation model (ALOS PALSAR RTC) with a resolution of 12.5 m was used, an Alaska Satellite Facility (ASF) product, in the Geophysical Institute at the University of Alaska Fairbanks. This model was developed by processing the synthetic aperture radar (SAR) data related to the Advanced Land Observing Satellite (ALOS). The Japan Aerospace Exploration Agency (JAXA) are responsible for this satellite. It is also known as DAICHI and was operated in an iterative cycle of 46 days from its launch in 2006 until 2011. The ASF made the data available for the users of geographical information systems in the form of Geo TIFF. The most crucial process is the radiographic and topographic correction as up-sampling is carried out on the cells of terrain data up to 12.5 m and by converting the resulting value of heights from orthometric height to ellipsoid height. These processed products are known as radiometric terrain correction (RTC). The quality of the ALOS PALSAR RTC is directly linked to the quality of the digital elevation model (DEM) used in the RTC. In the correction processes, a set of national elevation datasets (NED) was used with a resolution of 10 m, 30 m, and 60 m, in addition to the data of the Shuttle Radar Topography Mission (SRTM) with a resolution of 30 m. The SRTM covers the study area's range [40–42]. The DEM which covers the study area was obtained from the ASF Data Search Vertex [40], allowing free access to the remote sensing data.

#### 4.1.2. Cartographic Data

Nine topographic maps, scale 1:50,000 in Figure 2, cover the study area as follows: ABAR MUSIDDAH 24-4625 [43], Faydat Al Khafs Al Janubiyah 31-4625 [44], ATH THU-MAMAH 23-4625 [45], MALHAM 32-4625 [46], HURAYMILA 33-4625 [47], RAGHABAH

22-4525 [48], AL UYAYNAH 41-4624 [49], SUDUS 44-4624 [50], and AL BARRAH, 11-4524 [51]. The Ministry of Petroleum and Mineral Resources, Aerial Survey Department (1982) produced these maps—source: Saudi Geological Survey.

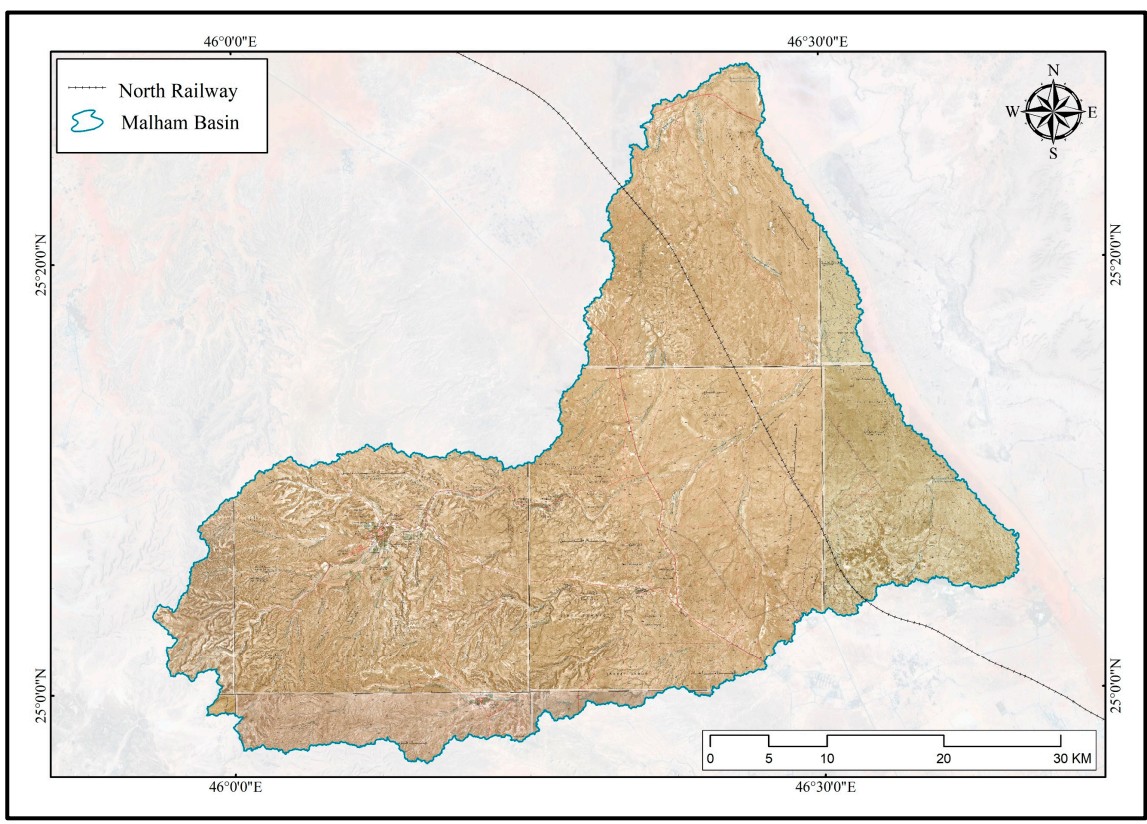

**Figure 2.** Topographic maps, scale: 1:50,000. Source: Ministry of Petroleum and Mineral Resources, 1982.

A total of 18 topographic maps, scale: 1:25,000 in Figure 3, covering some parts of the study area are as follows: KING KHALID WILDLIFE RESEARCH CENTRE 1-23-4625 [52], IRQ BANBAN 2-23-4625 [53], TUWAYQ TRADITIONAL VILLAGE 3-23-4625 [54], ATH THUMAMAH 4-23-4625 [55], SHAIB MALHAM 1-32-4625 [56], SULTANAH 2-32-4625 [57], SULBUKH 3-32-4625 [58], MALHAM 4-32-4625 [59], AL QIRINAH 1-33-4625 [60]), (JABAL AL ABRAQ 2-33-4625 [61]), HURAYMILA 3-33-4625 [62], JABAL ABA AL IDHAM 4-33-4625 [63], SUDUS 1-44-4624 [64], QARAT ABA AL GHIBTAN 4-44-4624 [65], AL UYAYNAH 1-41-4624 [66], BUDAH 4-41-4624 [67], DHAHRAT-MURIHAH 1-22-4525 [68], and JABAL AL HUSAYYINAT 2-22-4525 [69]. The General Commission created these maps for a survey (2015), source: General Authority for Survey and Geospatial Information. Both sets of maps (Figures 2 and 3) were used to ascertain the drainage network's path and the valley mouth's location.

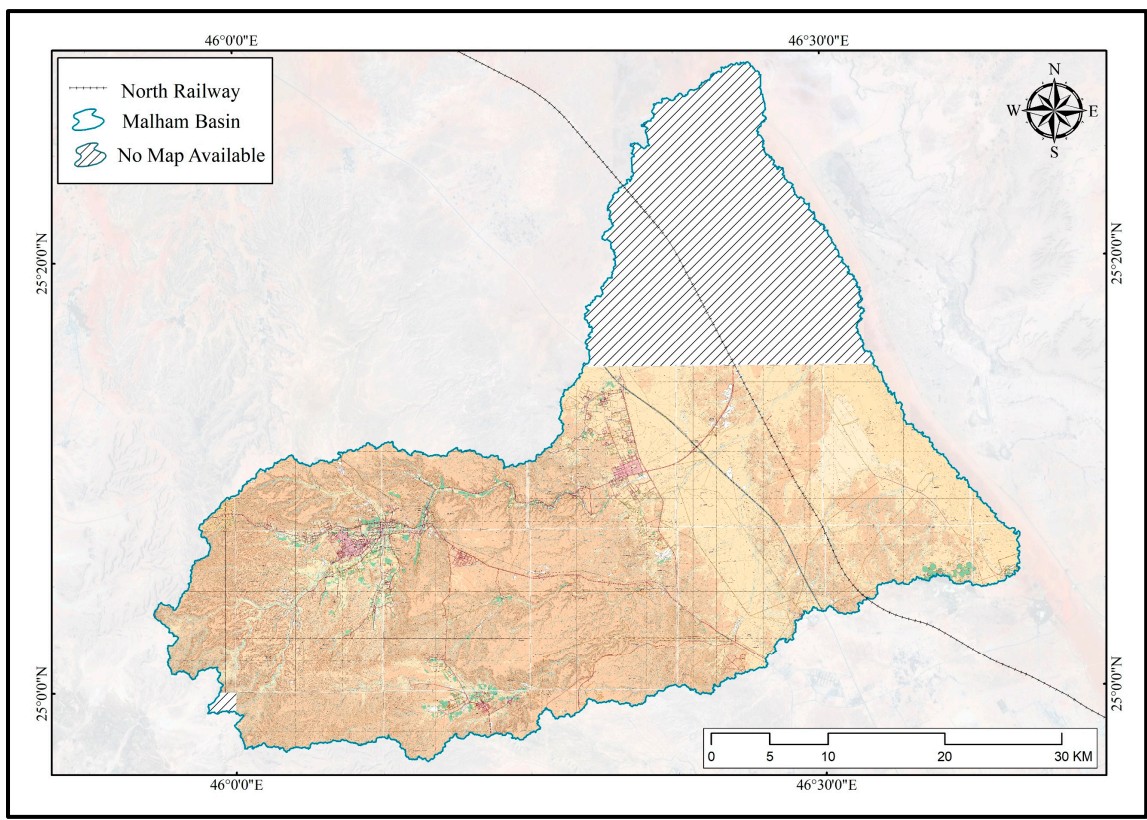

**Figure 3.** Topographic maps, scale: 1:25,000. Source: General Commission for Survey, 2015.

## 5. Approach of Study

This study followed the quantitative analytical approach in studying drainage basins. This approach depends on quantitative numbers and measurements in studying the phenomenon [70]. In geomorphological studies, this approach was developed by many scientists, the most famous of them being Horton and Strahler, to depend on various methods and tools of analysis, such as morphometric analysis [71,72]. This quantitative analytical approach was applied in this study through quantitative data analysis to study the characteristics of basins and morphometric drainage networks, which contributed to the classification of the hazard degree of basins.

### 5.1. Procedures of Study and the Most Important Phases

The procedures of the study and its most essential phases began from collecting and processing data, then analyzing and measuring morphometric parameters, and reaching the classification of hazard of basins, as clarified in the progress plan of the essential phases of the study in Figure 4.

### 5.2. Data Collection and Processing

This is the phase where data were collected from the abovementioned sources. Then, these data were processed ready for analysis through software, including: removing gaps from DEM, treatment of sewages, determining coordinates, georeferencing maps, and truncating maps and layers based on the boundaries of the study area.

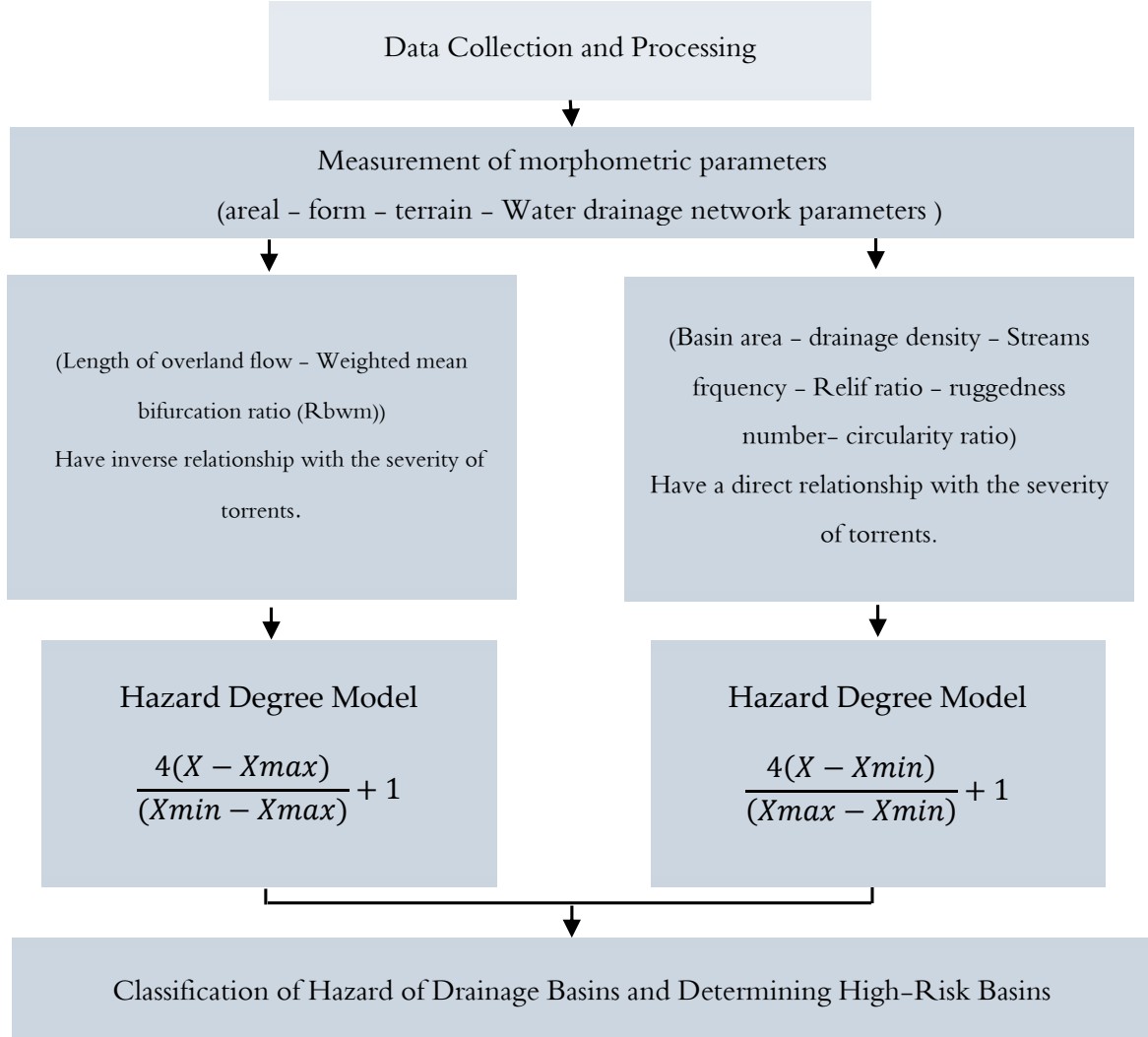

**Figure 4.** Study Methodology.

*5.3. Measurement of the Hazard Degree of the Direct Surface Runoff and Classification of Basins*

This was implemented by applying the quantitative analysis processes on the DEM to obtain different morphometric measurements (Table 1). These measurements can be used to measure and classify the hazard degree of drainage basins affecting the North Train Railway within Wadi Malham by applying the hazard degree model. Then, by using the cartographic method, the results were represented in the form of maps. Many studies, such as [31,73–75], referred to different morphometric measurements to determine the hazard degree of the direct surface runoff in drainage basins. Adnan et al. [76] stated that, based on the literature review, there is a consensus agreement on the specific morphometric measurements to use for defining areas of direct surface runoff hazard in drainage basins. This study approved eight parameters (length of overland flow, weighted mean bifurcation ratio ($R_{bwm}$), basin area, drainage density, stream frequency, relief ratio, ruggedness number, and circularity ratio) to measure the hazard, as clarified in Figure 4.

**Table 1.** Morphometric parameters used in applying the hazard degree model.

| Morphometric Measurement | Equation | Source of Equation |
|---|---|---|
| Basin area (A) | Measurement through ArcMap 10.7 | ([77]; documented in [17]) |
| Basin perimeter (P) | Measurement through ArcMap 10.7 | ([77]; documented in [17]) |
| Basin length ($L_b$) | Measurement through ArcMap 10.7 | ([77]; documented in [17]) |
| Stream order (u) | Hierarchical order | ([78]; documented in [79]) |
| No. of streams (Nu) | $N_u = N_1 + N_2 + \ldots + N_n$ | ([80]; documented in [79]) |
| Stream lengths ($L_u$) | $Lu = L_1 + L_2 + \ldots + L_n$ | ([81]; documented in [79]) |
| Drainage density ($D_d$) | $D_d = L_u/A$ <br> $L_u$ = Total stream lengths for all orders, A = Basin area | ([80]; documented in [17]) |
| Stream frequency ($F_s$) | $F_s = N_u/A$ <br> $N_u$ = Total no. of streams for all orders, A = Basin area | ([80]; documented in [17]) |
| Basin relief ($B_h$) | $B_h = h - h_1$ <br> Maximum height = h <br> Minimum height = $h_1$ | ([82]; documented in [83]) |
| Relief ratio ($R_r$) | $R_r = B_h/L_b$ <br> $B_h$ = Basin relief, $L_b$ = Basin length | ([84]; documented in [83]) |
| Ruggedness number ($R_n$) | $R_n = D_d \times (B_h/1000)$ <br> $B_h$ = Basin relief, $D_d$ = drainage density | ([81]; documented in [81]) |
| Circularity ratio ($R_c$) | $R_c = 4\pi A/P^2$ <br> A = Basin area, $\pi$ = 3.1415, <br> $P^2$ = Basin perimeter square | ([85]; documented in [17]) |
| Length of overland flow ($L_o$) | $1/2D_d$ <br> $D_d$ = Drainage density | ([80]; documented in [17]) |
| Weighted mean bifurcation ratio (Rbwm) | $Rbwm = \sum(Rb \times Nu - r)/\sum Nu - r$ <br> Rb= Bifurcation ratio (Nu/Nu + 1), Nu − r = Total amount of <br> drainage used in measuring the bifurcation ratio <br> (Nu + (Nu + 1)) | ([86]; documented in [75]) |

Concerning this model, Bajabaa et al. [31] explained that some morphometric parameters have hydrological indications and impact the direct surface runoff in drainage basins. If the morphometric coefficient has a direct relationship with the severity of the direct surface runoff in the basin, the hazard degree is measured by Equation (1) ([87]; documented in [31]).

$$Hazard\,degree = \frac{4(X - Xmin)}{(Xmax - Xmin)} + 1 \qquad (1)$$

If the morphometric coefficient has an inverse relation with the severity of direct surface runoff, in this case, the hazard degree is measured by Equation (2) (Davis, 1975; documented in [31]).

$$Hazard\,degree = \frac{4(X - Xmax)}{(Xmin - Xmax)} + 1 \qquad (2)$$

*X* is the morphometric coefficient value used to evaluate each basin's hazard degree. *Xmax* is the maximum value among the morphometric parameters of all basins. *Xmin* is the minimum value among the morphometric parameters of all basins. After applying the two equations, the morphometric hazard degree measured for all basins is added separately to reach the gross value of the hazard degree of basins. These basins are classified into selected categories accordingly. The measurement of hazard degree ranges from 1 (minimum) to 5 (maximum); i.e., the higher the value, the greater the hazard.

## 6. Analysis and Results

First: Calculation of Hazard Degree on the Direct Surface Runoff in Drainage Basins and its Classification

### 6.1. Morphometric Parameters and Their Hydrological Indications

The drainage network is derived from the DEM ALOS PALSAR RTC [40], Figure 5, with a sensing value of 100. This value represents the accumulation of runoff; then, the water divide lines are determined, and streams of basins affecting the North Train Railway within Wadi Malham are treated in Figure 6, using ArcMap 10.7. The topographic maps in Figures 2 and 3 verify the drainage networks derived from the DEM and identify the topography of the study area, upstream area, and designations of the valley. This is followed by calculating many morphometric parameters in Table 2 to measure the hazard degree of direct surface runoff in drainage basins and its classification by applying the hazard degree model.

**Table 2.** Morphometric parameters of drainage basins and networks in the study area.

| Parameters | Basin No. | | | | | | | | | | | | | |
|---|---|---|---|---|---|---|---|---|---|---|---|---|---|---|
| | 1 | 2 | 3 | 4 | 5 | 6 | 7 | 8 | 9 | 10 | 11 | 12 | 13 | 14 |
| Basin area (km$^2$) | 8.87 | 154.61 | 7.74 | 7.10 | 4.78 | 5.57 | 5.76 | 1137.37 | 8.43 | 3.55 | 8.24 | 14.57 | 14.92 | 4.43 |
| Basin perimeter (km) | 17.68 | 84.46 | 16.75 | 14.82 | 14.24 | 13.03 | 13.27 | 213.49 | 19.22 | 11.95 | 16.68 | 25.15 | 20.65 | 12.48 |
| Basin length (km) | 5.39 | 23.62 | 3.73 | 3.60 | 3.90 | 4.42 | 4.32 | 55.36 | 5.50 | 3.20 | 4.21 | 6.83 | 5.10 | 3.40 |
| Steam order | 4 | 7 | 5 | 5 | 4 | 4 | 4. | 8 | 4 | 4 | 5 | 5 | 5 | 5 |
| No. of streams (stream) | 205 | 3521 | 174 | 162 | 114 | 130 | 143 | 25441 | 178 | 90 | 191 | 352 | 356 | 110 |
| Stream length (km) | 51.15 | 928.05 | 44.26 | 41.86 | 27.78 | 34.42 | 34.19 | 6327.95 | 47.89 | 19.95 | 48.26 | 83.67 | 89.93 | 27.86 |
| Drainage density (km$^2$/km) | 5.76 | 6.00 | 5.72 | 5.90 | 5.81 | 6.18 | 5.93 | 5.56 | 5.68 | 5.62 | 5.86 | 5.74 | 6.03 | 6.29 |
| Stream frequency (streams/km$^2$) | 23.10 | 22.77 | 22.48 | 22.81 | 23.85 | 23.33 | 24.81 | 22.37 | 21.10 | 25.36 | 23.17 | 24.15 | 23.86 | 24.85 |
| Highest point level (m) | 634 | 755 | 641 | 640 | 651 | 661 | 663 | 981 | 665 | 668 | 678 | 704 | 725 | 705 |
| Lowest point level (m) | 598 | 601 | 609 | 616 | 617 | 624 | 625 | 624 | 626 | 640 | 642 | 641 | 653 | 661 |
| Basin relief (m) | 36 | 154 | 32 | 24 | 34 | 37 | 38 | 357 | 39 | 28 | 36 | 63 | 72 | 44 |
| Relief ratio | 6.68 | 6.52 | 8.59 | 6.66 | 8.72 | 8.37 | 8.81 | 6.45 | 7.09 | 8.76 | 8.54 | 9.23 | 14.11 | 12.94 |
| Ruggedness number | 0.21 | 0.92 | 0.18 | 0.14 | 0.20 | 0.23 | 0.23 | 1.99 | 0.22 | 0.16 | 0.21 | 0.36 | 0.43 | 0.28 |
| Circularity ratio | 0.36 | 0.27 | 0.35 | 0.41 | 0.30 | 0.41 | 0.41 | 0.31 | 0.29 | 0.31 | 0.37 | 0.29 | 0.44 | 0.36 |
| Length of overland flow (km) | 0.0867 | 0.0833 | 0.0875 | 0.0848 | 0.0861 | 0.0809 | 0.0843 | 0.0899 | 0.0881 | 0.0889 | 0.0854 | 0.0871 | 0.0829 | 0.0795 |
| Weighted mean bifurcation ratio | 4.00 | 4.16 | 4.25 | 4.05 | 3.97 | 3.93 | 3.92 | 4.08 | 4.69 | 4.24 | 3.92 | 3.81 | 3.97 | 3.91 |

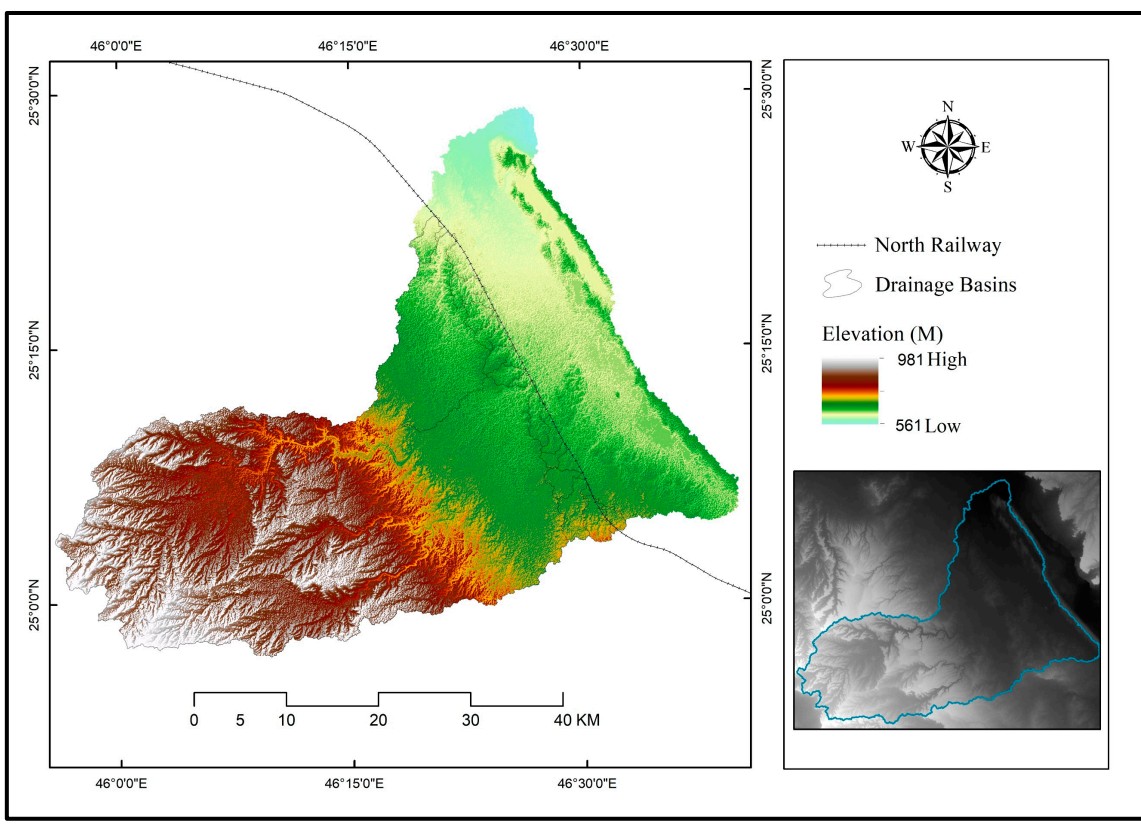

**Figure 5.** DEM ALOS PALSAR RTC for the study area. Source: [40].

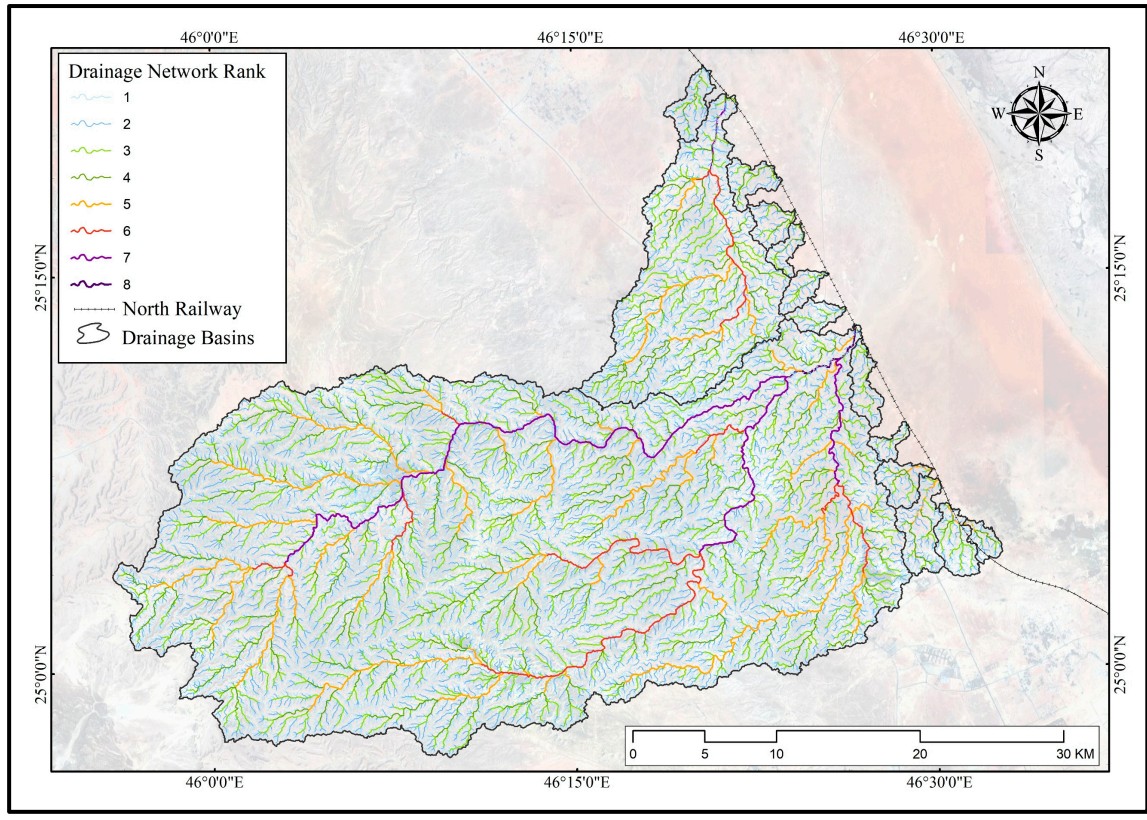

**Figure 6.** Drainage basins and networks affecting the North Train Railway within the basin of Wadi Malham.

### 6.2. Basin Area (A)

This is the measurement of the confined surface in the water divide line. It is calculated in this study in km$^2$, through the method of calculating geometry in ArcMap. As mentioned, different opinions exist on the relevance between the basin area and the top of the direct surface runoff drainage [17]. In this study, the standard of the basin area is considered to indicate the basin size and describes the relationship as direct between them; as a result, the principle of the larger the area of the drainage basins, the greater their chance of receiving and collecting rainwater was followed. Thus, the volume of the peak flow increases with the increase in the amount that falls into the basin [4,17,88,89]. This means that the hazard of direct surface runoff increases in the case of an increase in the areas of drainage basins. Through the measurement, the basin area in the study area ranges from 3.55 km$^2$ to 1137.37 km$^2$, as shown in Table 2 and Figure 7.

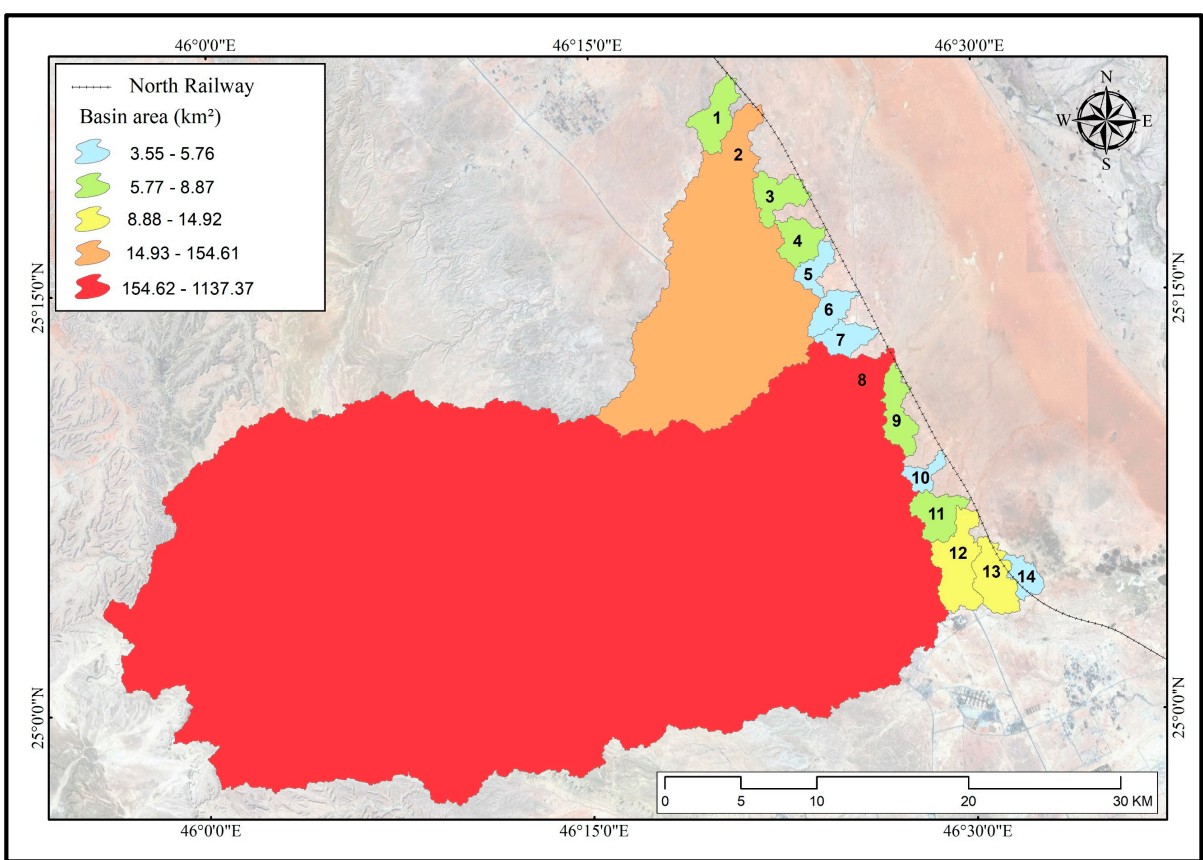

**Figure 7.** Basin area.

### 6.3. ($L_b$) Basin Length

This represents the maximum basin length. The distance is measured in parallel with the primary runoff from the mouth to the highest point in the basin upstream [78], documented in [17]. In this study, it was measured by the measurement tools in ArcMap 10.7, with the unit of measurement of km. The lengths of drainage basins in the study area range from 3.20 km to 55.36 km, as shown in Table 2.

### 6.4. Basin Perimeter (P)

This measures the length of the water divide line, which separates the basins from their neighbors [5]. It was measured in this study in km through calculating geometry in ArcMap 10.7. The measurements of basin perimeters in the study area range from 11.95 km to 213.49 km, as shown in Table 2.

### 6.5. Number of Streams ( $N_u$ ) and Stream Order (u)

Al-Maghazi [74] mentioned in his study that one of the methods for the classification of stream order is the method proposed by Strahler [78], which depends on considering the streams which are not connected to any previous stream as first-order streams. This method was applied in this study when classifying the stream orders through the tool stream to order. If two streams of the same order are connected, the following stream shall have a higher order. The second order arises from the connection of two first-order streams, and so on for the remaining orders. In the case of connection of two streams from different orders, the following stream shall have the same order as the stream that is higher in level. When describing the basin order, it is based on the higher order which pours into its exit. The number of streams for each order is calculated through the tool summary statistics, found in ArcMap 10.7. It is evident in Tables 2 and 3 that the classification of basin orders in the study area is variable between a fourth-order basin, where the total number of streams is 90, and an eighth-order basin, where the total number of basins is 25,441.

**Table 3.** Numbers and orders of streams in drainage basins.

| Basin No. | Order | | | | | | | | Total |
|---|---|---|---|---|---|---|---|---|---|
| | First | Second | Third | Fourth | Fifth | Sixth | Seventh | Eighth | |
| 1 | 148 | 49 | 7 | 1 | - | - | - | - | 205 |
| 2 | 2674 | 652 | 150 | 35 | 7 | 2 | 1 | - | 3521 |
| 3 | 133 | 31 | 7 | 2 | 1 | - | - | - | 174 |
| 4 | 122 | 30 | 7 | 2 | 1 | - | - | - | 162 |
| 5 | 84 | 25 | 4 | 1 | - | - | - | - | 114 |
| 6 | 96 | 28 | 5 | 1 | - | - | - | - | 130 |
| 7 | 106 | 30 | 6 | 1 | - | - | - | - | 143 |
| 8 | 19,173 | 4800 | 1194 | 209 | 52 | 9 | 3 | 1 | 25,441 |
| 9 | 140 | 30 | 7 | 1 | - | - | - | - | 178 |
| 10 | 69 | 16 | 4 | 1 | - | - | - | - | 90 |
| 11 | 142 | 37 | 8 | 3 | 1 | - | - | - | 191 |
| 12 | 259 | 72 | 17 | 3 | 1 | - | - | - | 352 |
| 13 | 264 | 74 | 13 | 4 | 1 | - | - | - | 356 |
| 14 | 81 | 22 | 4 | 2 | 1 | - | - | - | 110 |

### 6.6. Stream Lengths ($L_u$)

The total stream lengths of all orders in each basin are measured in km by calculating geometry in ArcMap 10.7. The minimum total length of streams is 19.95 km. On the other hand, the maximum total length of streams is 6327.95 km, as shown in Table 2.

### 6.7. ($D_d$) Drainage Density

This represents the total lengths of streams on the basis of each unit of area divided by the basin area ([80]; documented in [17]). This is measured in $km/km^2$, which indicates how small the distance between streams is and reflects the extent of discontinuity on the basin surface. Moreover, it gives an idea of the permeability of the rocks, as high drainage density values indicate the severity of the basin's surface, high rates of direct surface runoff, and a reduction in leakage ratios. Moreover, the opposite occurs concerning lower values. Thus, it directly relates to the hazard of direct surface runoff [17,90]. The drainage density of basins in the study area ranges from 5.56 $km/km^2$ to 6.29 $km/km^2$, as shown in Table 2 and Figure 8. Perhaps, these high values are due to the nature of the area and type of rocks, which have low permeability.

### 6.8. ($F_s$) Stream Frequency

This coefficient measures the ratio of the number of streams in basins to their area ([80]; documented in [17]) in streams/$km^2$. Values of stream frequency in basins of the study area, as shown in Table 2 and Figure 9, range from 21.10 streams/$km^2$ to 25.36 streams/$km^2$.

These are high values, and as mentioned by the authors of [91], this is due to the high accuracy of upstream areas, from which the streams are derived. This leads to the possibility of confinement of most of the streams in the study area. In addition, this is also due to the significant impact of the type of rocks on the plentifulness of streams. Stream frequency and direct surface runoff have a direct relationship because a high stream frequency in a basin indicates low permeability of the surface of the basin and low leakage rates, resulting in an increase in the possibility of stronger surface runoff compared to basins of low frequency [17,90].

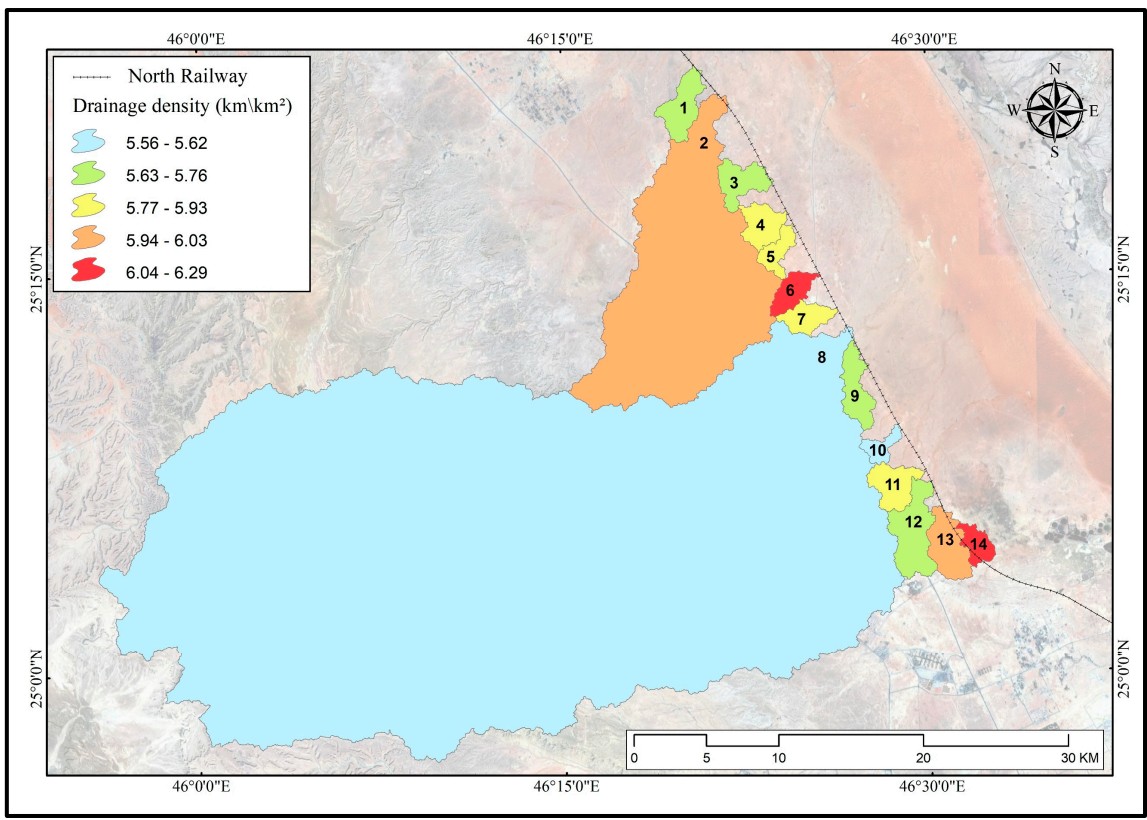

**Figure 8.** Drainage density in basins.

### 6.9. ($B_h$) Basin Relief

This is also called total relief. Its equation is proposed by the authors of [77] and it is measured by finding the difference between the highest point level and the lowest point level in the drainage basin (documented in [83]). The basin relief in this study is measured in meters using the values registered in the DEM. The results in Table 2 show that the basin reliefs of the study area range from 24 m to 357 m.

### 6.10. ($R_r$) Relief Ratio

This is the ratio without any dimensions which measures the relationship between the relief and length of the basin) [77]; documented in [83]). It rises in the drainage basin by the increase in the difference between the highest and lowest point levels. It decreases with the increase in the basin length. It reflects the inclination degree of the basin surface. It gives an idea of how fast the direct surface runoff is. Its rise in drainage basins indicates severe inclinations and an increase in velocity of the direct surface runoff; this also indicates an increase in hazard in the drainage basins. Thus, they have a direct relationship [9,74,83]. It is clear from the calculations, as shown in Table 2 and Figure 10, that the relief ratio ranges from 6.45 to 14.11 in drainage basins.

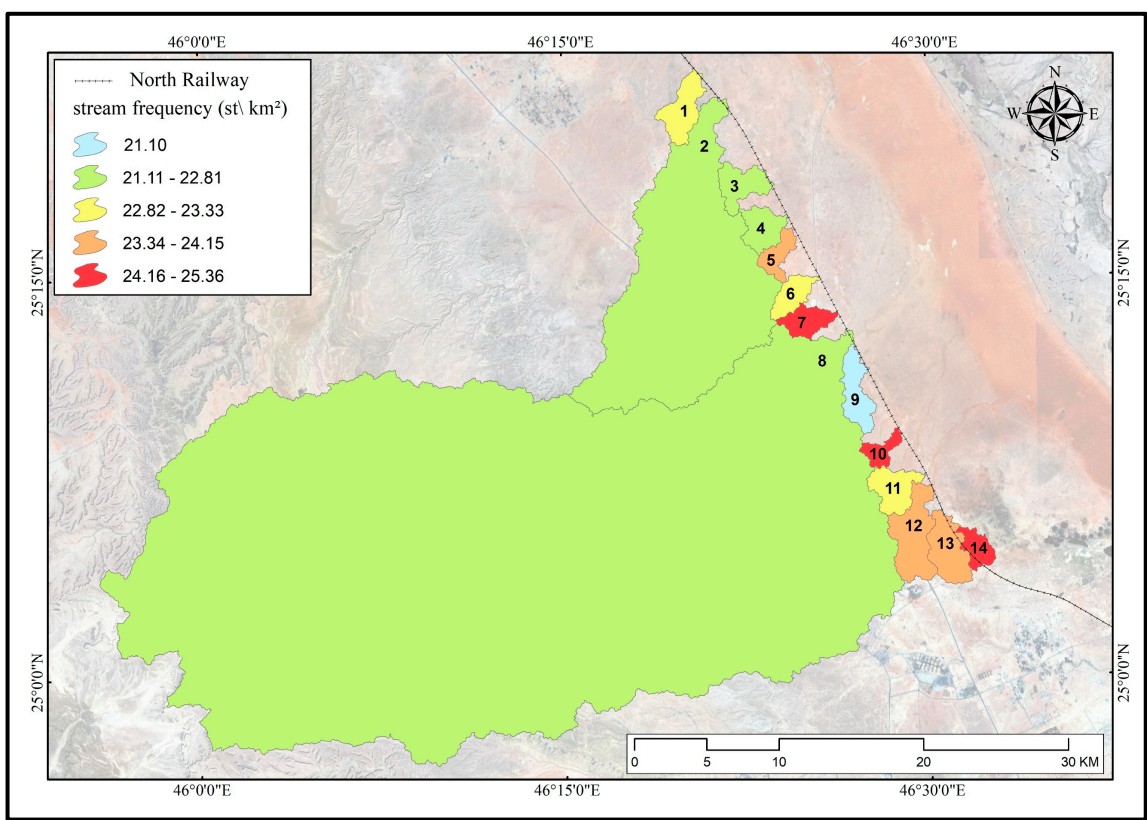

**Figure 9.** Stream frequency in basins.

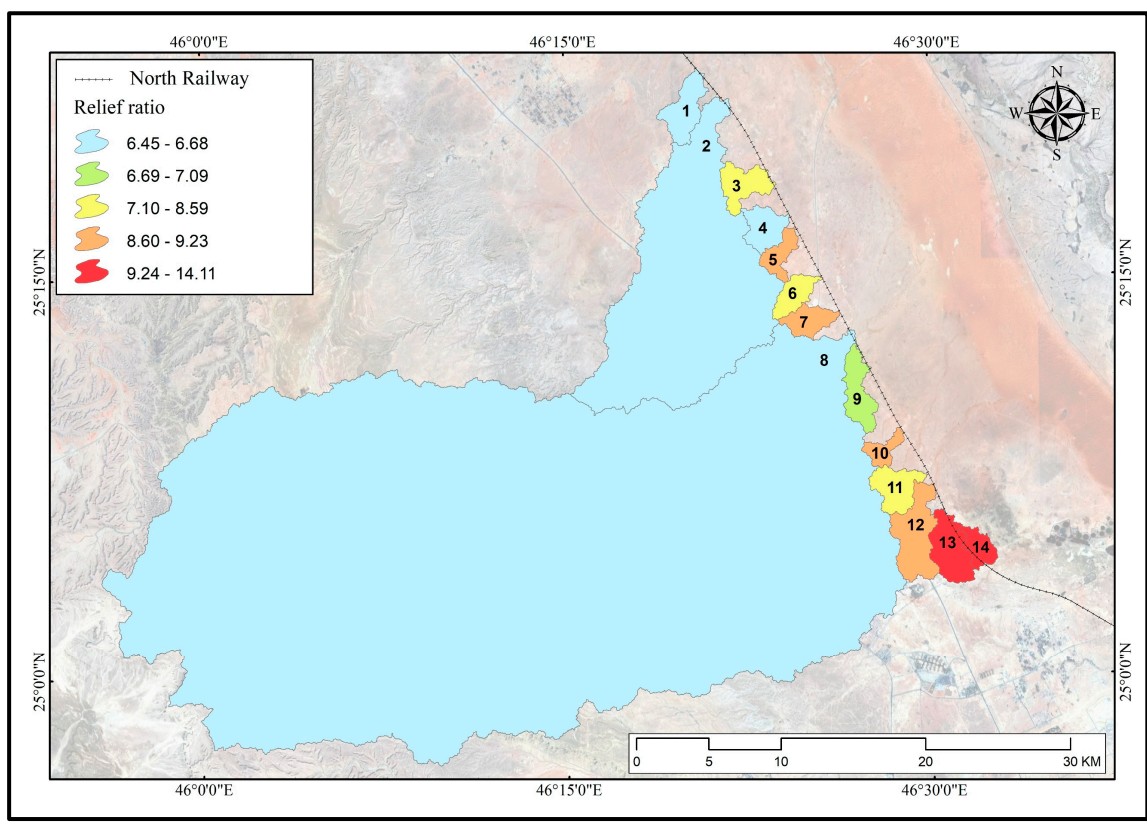

**Figure 10.** Basin relief ratio.

### 6.11. ($R_n$) Ruggedness Number

This is a coefficient without dimensions which measures the relationship between the basin relief and drainage density ([81,92]; documented in [83]). Strahler [81] states that the rise in the values of drainage density and basin relief values leads to a rise in the ruggedness number (documented by the authors of [93]). High ruggedness numbers in basins indicate severe and long inclinations [83]. This increases the possibility of generating direct surface runoff of high peak flows compared to basins of low ruggedness ([94]; documented in [95]), i.e., there is a direct relationship between them. The ruggedness number, as shown in Table 2 and Figure 11, ranges from 0.14 to 1.99 in basins.

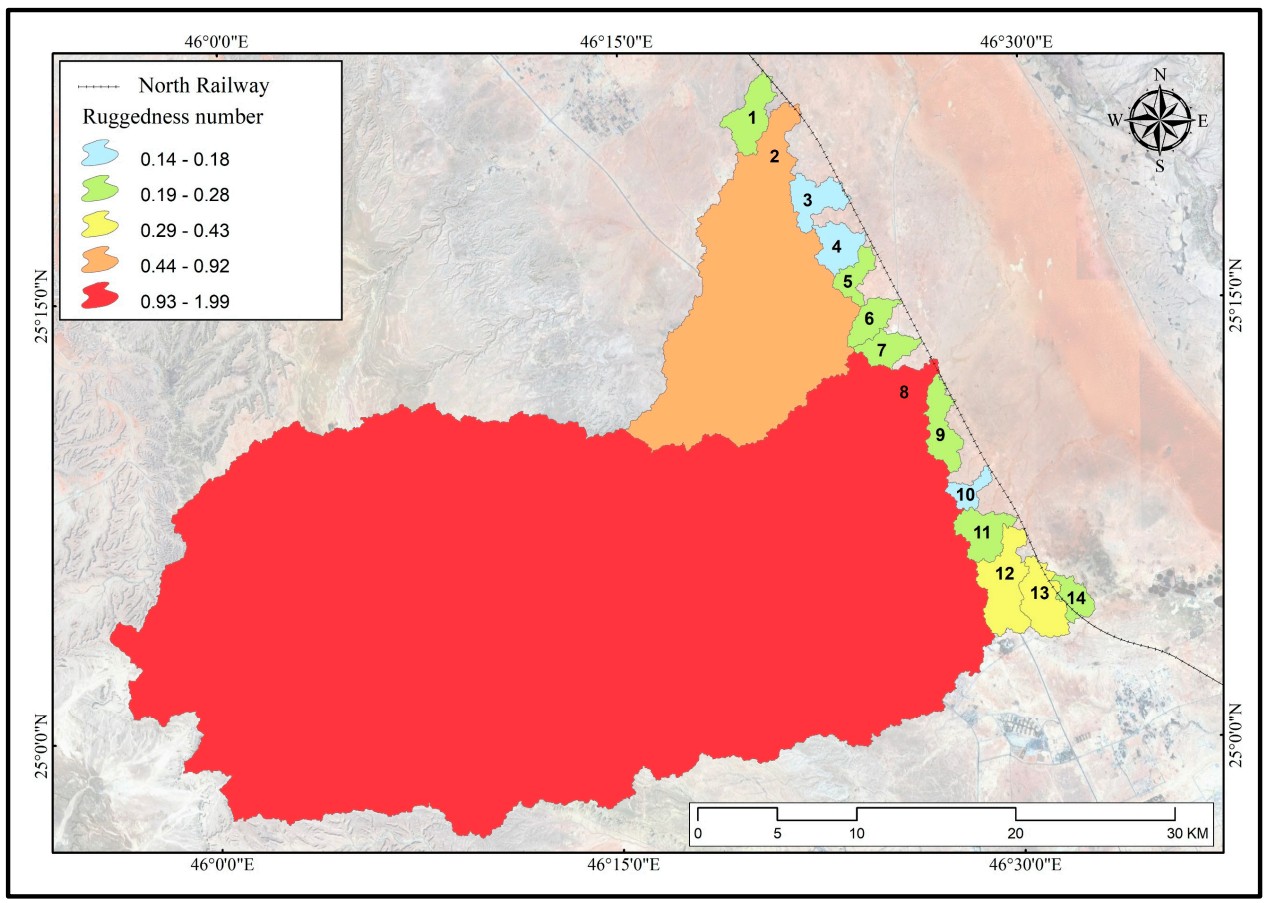

**Figure 11.** Ruggedness number of basins.

### 6.12. ($R_c$) Circularity Ratio

This is the basin area ratio to the circle area, which has the same perimeter as the basin ([85]; documented in [17]). It is an indicator that gives an idea of the form of the basin, the quantity of direct surface runoff, and the time needed to reach the exit of the basin. The circularity ratio is represented in values from 0 to 1. High values, which are close to 1, indicate that the form of the drainage basin is more like a regular circle; however, if the values are closer to 0, this indicates the increase in extension and elongation of the basin. This affects the nature of direct surface runoff as the running water in more circular basins arrives all at once with a high velocity and peak compared to the less circular basins [17,93]. Based on the above, the circularity ratio of the basin has a direct relationship with the hazard degree of direct surface runoff. Concerning the drainage basins in the study area, the circularity ratio in all drainage basins refers to the fact that the basins are known to be extended rather than circular. The ratios in Table 2 and Figure 12 range between 0.27 and 0.44.

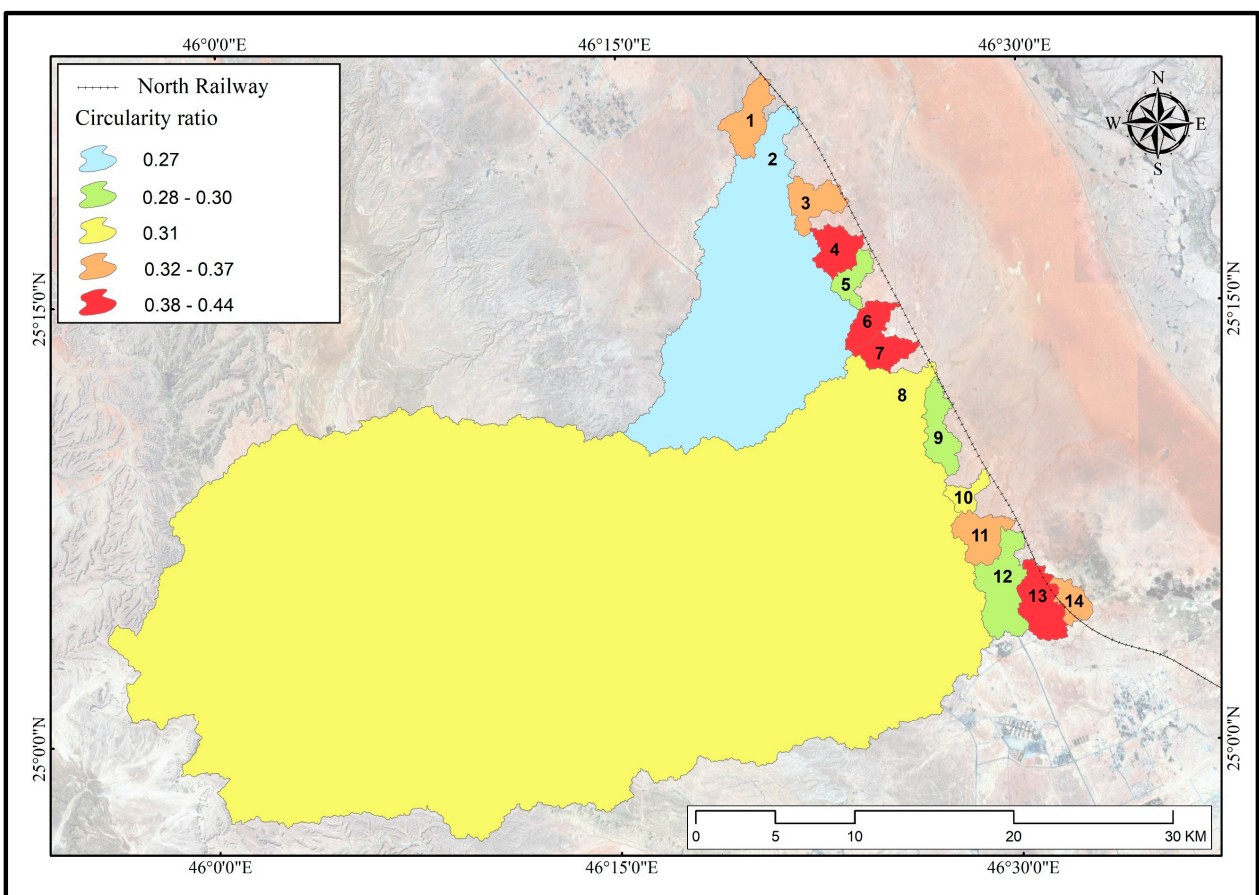

**Figure 12.** Circularity ratio of basins.

### 6.13. Length of Overland Flow ($L_o$)

This measure is the length of overland flow before it accumulates in channels of a particular stream ([80]; documented in [96]). Low values in drainage basins indicate that the water is concentrated more quickly in streams compared to basins of high values. The relationship between the overland flow length and the direct surface runoff hazard degree is inverse [17]. By calculating the length of the overland flow of drainage basins in the study area, it is found that the values range from 0.0795 km to 0.0899 km, as shown in Table 2 and Figure 13.

### 6.14. Weighted Mean Bifurcation Ratio

$R_{bmw}$ is proposed by the author of [86] as a modification to the method of calculating the $B_R$ as he believes that it is more accurate (documented in [17]). It is obtained by multiplying the Rb of orders by the total number of streams used in measuring the Rb to all orders in the basin; then, the total average of these values is calculated [75]. Its high values refer to the basins that are severely extended and discontinuous in streams and where the water scatters. It takes longer before reaching the main stream. They also refer to the direct surface runoff flowing slowly in the drainage basin. Therefore, the Rb has an inverse relationship with the hazard degree of the direct surface runoff [32,89,93]. The Rb in drainage basins in the study areas ranges from 3.81 to 4.69, as shown in Table 2 and Figure 14.

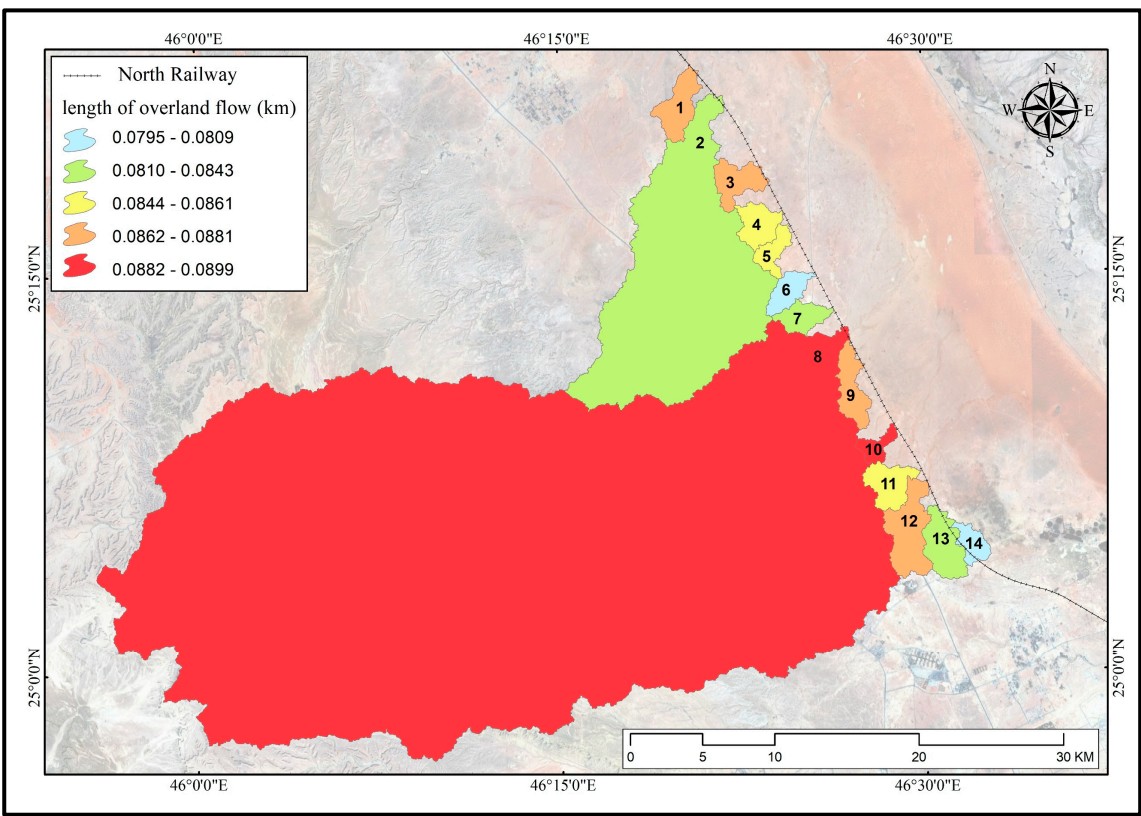

**Figure 13.** Length of overland flow in basins.

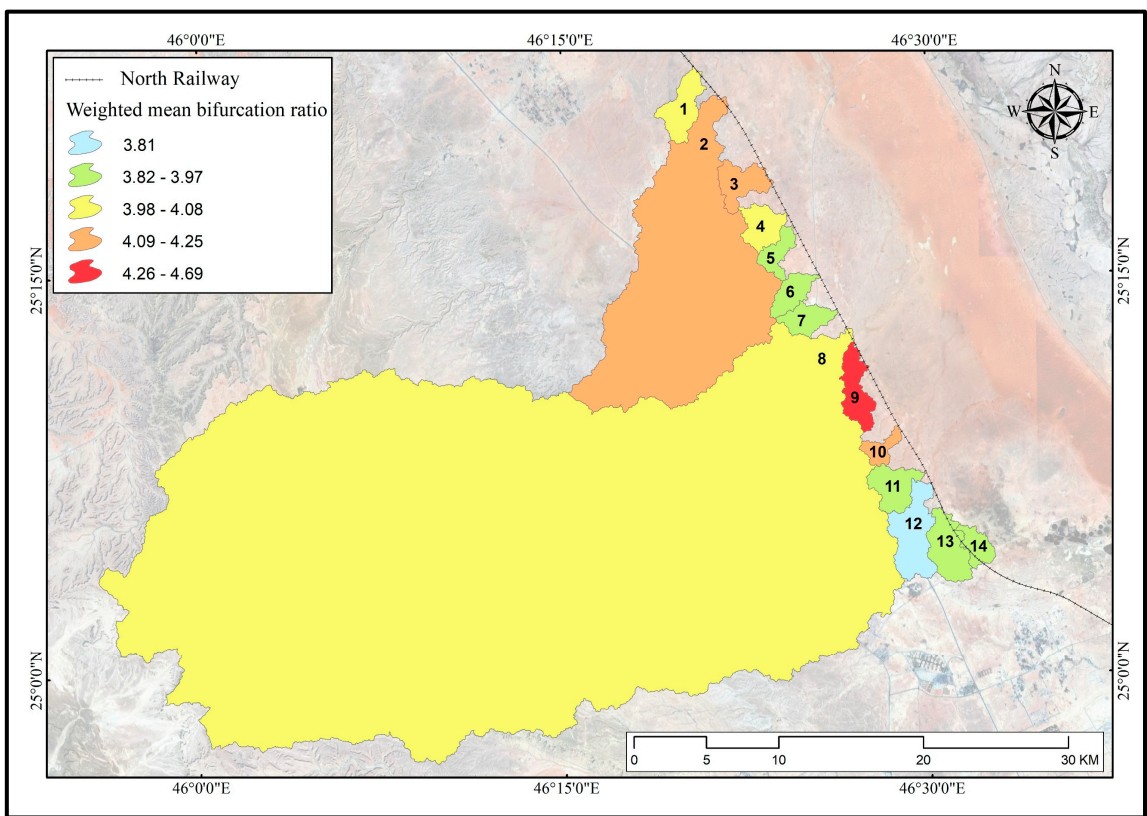

**Figure 14.** Weighted mean bifurcation ratio in drainage basins.

### 6.15. Application of the Hazard Degree Model

The hazard degree model is applied through Equations (1) and (2) on eight parameters with hydrological indications, giving an idea of the behavior of direct surface runoff. It interferes with the increase and decrease in the hazard of direct surface runoff. They are shown in Table 4, which also shows the results of applying this model to drainage basins in the study area.

**Table 4.** Results of applying the hazard degree model to drainage basins in the study area.

| Parameters | Basin No. | | | | | | | | | | | | | |
|---|---|---|---|---|---|---|---|---|---|---|---|---|---|---|
| | 1 | 2 | 3 | 4 | 5 | 6 | 7 | 8 | 9 | 10 | 11 | 12 | 13 | 14 |
| Basin area (km$^2$) | 1.0188 | 1.5329 | 1.0148 | 1.0125 | 1.0043 | 1.0071 | 1.0078 | 5 | 1.0172 | 1 | 1.0166 | 1.0389 | 1.0401 | 1.0031 |
| Drainage density (km$^2$/km) | 2.0991 | 3.4078 | 1.8437 | 2.8188 | 2.3511 | 4.3642 | 3.0179 | 1 | 1.6243 | 1.3187 | 2.5990 | 1.9751 | 3.5492 | 5 |
| Stream frequency (streams/km$^2$) | 2.8764 | 2.5696 | 2.2904 | 2.6058 | 3.5781 | 3.0951 | 4.4853 | 2.1890 | 1 | 5 | 2.9455 | 3.8671 | 3.5937 | 4.5183 |
| Relief ratio | 1.1216 | 1.0378 | 2.1165 | 1.1116 | 2.1865 | 2.0018 | 2.2302 | 1 | 1.3348 | 2.2045 | 2.0927 | 2.4496 | 5 | 4.3870 |
| Ruggedness number | 1.1432 | 2.6976 | 1.0899 | 1 | 1.1215 | 1.1888 | 1.1819 | 5 | 1.1733 | 1.0345 | 1.1503 | 1.4775 | 1.6343 | 1.2936 |
| Circularity ratio | 3.0195 | 1 | 2.7794 | 4.2075 | 1.5708 | 4.3495 | 4.3186 | 1.9860 | 1.3516 | 1.9527 | 3.3847 | 1.4088 | 5 | 3.0295 |
| Length of overland flow (km) | 2.1999 | 3.5242 | 1.9285 | 2.9414 | 2.4634 | 4.4273 | 3.1408 | 1 | 1.6920 | 1.3567 | 2.7184 | 2.0688 | 3.6610 | 5 |
| Weighted mean bifurcation ratio | 4.1621 | 3.4131 | 3.0042 | 3.9128 | 4.2891 | 4.4831 | 4.4889 | 3.7865 | 1 | 3.0410 | 4.5192 | 5 | 4.2644 | 4.5349 |
| Total hazard degree | 17.641 | 19.183 | 16.067 | 19.610 | 18.565 | 24.917 | 23.871 | 20.961 | 10.193 | 16.908 | 20.426 | 19.286 | 27.743 | 28.766 |

Drainage basins are classified into three categories of hazards, as applied in some previous studies, such as [31,97]. The length of the category was determined through Equation (3) [4] by subtracting the lowest value from the highest value of total hazard degrees of morphometric parameters; then, it was divided by the number of categories (3), and the result was 6.191.

$$\text{CategoryLength} = \frac{\sum \text{Nmax} - \sum \text{Nmin}}{n} \tag{3}$$

Then, the resulting length of the category was added to the lowest value of total hazard degree of morphometric parameters to represent the highest limit of the first category. Then, the length of the category was added to each high limit to represent the beginning of the following category limit. The low category is represented by hazard degree (1); the higher category is represented by hazard degree (5); between them is the moderate hazard degree, which is represented by (3), as shown in Table 5.

**Table 5.** Categories of hazard degrees.

| Range | Measurement of Hazard Degree | Hazard Degree |
|---|---|---|
| 10.193–16.384 | 1 | Low |
| 16.384–22.575 | 3 | Moderate |
| 22.575–28.766 | 5 | High |

Based on the above, it is shown in Figure 15 that basins (3) and (9) have lower total values of hazard degree; thus, they fall within the low category of hazard degree. Regarding

basins 1, 2, 4, 5, 8, 10, 11, and 12, they fall within the moderate hazard category; finally, basins 6, 7, 13, and 14 fall within the high category of hazard. The basins in this category are subject to hydrological modeling and evaluation of sluices.

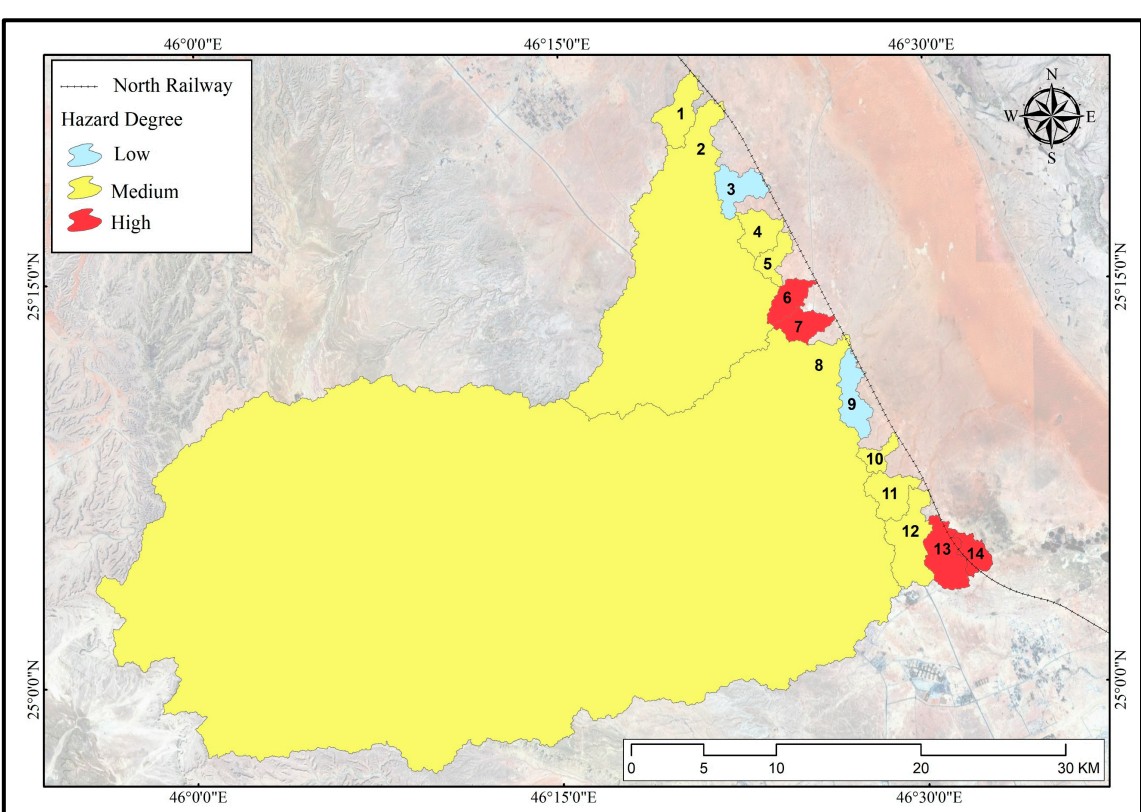

**Figure 15.** Hazard degree of direct surface runoff in drainage basins in the study area.

## 7. Discussion

Technical progress and the information revolution in geographic information systems and remote sensing have helped to significantly improve geomorphological studies and obtain better accuracy in terms of the results achieved. This study used spatial data and technologies employed using a scientific methodology to estimate the hazard of direct surface runoff on the North Train Railway, extended within the drainage basin of Wadi Malham. This is because accurate and modern scientific methods are used to study the basins, which contain essential strategic facilities, such as railways. It shall be determined whether or not there are places subject to direct surface runoff, which require priority in setting suitable policies and procedures to reduce its hazards.

The hazard degree model was applied by using six morphometric parameters that have a direct relationship with the hazard of direct surface runoff: basin area, drainage density, stream frequency, relief ratio, ruggedness number, and circularity ratio. This was in addition to parameters with an inverse relationship with the hazard of direct surface runoff: length of overland flow and weighted mean bifurcation ratio over the 14 basins in the study area. Drainage basins are classified into three categories of hazards, as has been applied in some previous studies such as [31,97]. According to the results, 14.29% of the total basins in the study area have the lowest overall values, 57.14% represent moderate-risk basins, and 28.57% are high-risk basins. The study concludes that two basins, 3 and 9, have low total values of hazard degree as they are equal to 10.193 and 16.067, respectively. They are classified in the category of low risk. The basins of moderate risk are ordered based on the total hazard degrees from the lowest to the highest: basins 10, 1, 5, 2, 12, 4, 11, and 8, and the total hazard degrees for each basin are equal to 16.908, 17.641, 18.565, 19.183, 19.286, 19.610, 20.426, and 20.961, respectively. Finally, basins 6, 7, 13, and 14 are considered to be

high risk, and the total hazard degree for each of these basins is 23.871, 24.917, 27.743, and 28.766, respectively. It can be noticed that the basin area did not affect the degree of hazard. This can be seen for basins 2 and 8, which are considered moderately risky, while their area is the largest compared with other basins.

Through Table 4, it becomes evident that basin 14 is the most hazardous basin with a total score of 28.766. The basin's area did not contribute significantly to its classification within the high-risk category, as it added only 1.003 degrees to the risk score. Regarding the ruggedness number, it contributed 1.29 degrees. As for the circularity ratio of the basin, it added 3.03 degrees to the total risk score. On the other hand, drainage density and the length of overland flow significantly increased the risk for this basin, adding 5 degrees each to the total risk score.

Following these factors are the weighted mean bifurcation ratio, stream frequency, and relief ratio, which added 4.54, 4.52, and 4.39 degrees to the total risk score, respectively. Basin 13 came next in terms of risk, with a total score of 27.743. Here, the basin's area did not contribute significantly, adding a risk score of 1.04. The ruggedness number contributed 1.63 degrees, and the drainage density contributed 3.55 degrees to the risk. The stream frequency added around 3.59 degrees, and the length of overland flow added 3.66 degrees. The weighted mean bifurcation ratio added 4.26 degrees, and the relief ratio added 5 degrees to the risk score, similar to the circularity ratio.

The table also reveals that basin 6 is the third most hazardous basin with a total score of 24.917. This is due to the significant contributions of the weighted mean bifurcation ratio (4.48 degrees), drainage density (4.36 degrees), circularity ratio (4.35 degrees), length of overland flow (4.43 degrees), and stream frequency (3.10 degrees). The relief ratio added a total of 2.00 degrees to the risk score, while the ruggedness number contributed approximately 1.18 degrees. The area parameter only contributed 1.00 degree.

As for basin 7, it ranks as the fourth most hazardous basin in the study area with a total score of approximately 23.871. This score resulted from the combined contributions of stream frequency and the weighted mean bifurcation ratio (4.49 degrees each), circularity ratio (4.32 degrees), length of overland flow (3.14 degrees), drainage density (3.02 degrees), relief ratio (2.23 degrees), ruggedness number (1.18), and, finally, the basin's area (1.01). All four of these basins in the study fall within the high-risk category. We conclude from this that the area did not contribute significantly to the risk scores.

## 8. Conclusions and Recommendations

Drainage basins affecting the North Train Railway within Wadi Malham, with an area of 2.59 km$^2$ or more, were studied due to the restrictions of unit hydrograph theory. We analyzed the degree of risk of 14 drainage basins affecting the North Train Railway in the study area. In light of the results of the study based on the applicable methodology and methods herein, the study showed, through a hazard degree map, the priorities of the drainage basins in terms of degrees of danger. This led to recommendations centered around better planning and management of railway infrastructure. The study recommends using the spatial database resulting from this study in various fields, such as the application of hydrological modeling by using unit hydrograph methods. This is because this method assists with understanding the characteristics of drainage basins and their networks. It is also the basis for some hydrological measurements that contribute to estimating the characteristics of direct surface runoff, as well as determining their importance in estimating its severity because of their hydrological indications [9]. Moreover, estimation of rainfall depth during different return periods, soil permeability analysis, and land use classification in the study area are recommended as the basis for future study. This is due to their role in generating direct surface runoff in drainage basins and thus in applying hydrological modeling and deriving a synthetic unit hydrograph for drainage basins. This contributes to the estimation of the volume and peak of the direct surface runoff in such arid and semi-arid environments that do not contain hydrometric stations to monitor the runoff. It is also recommended that the hydraulic efficiency of the existing drainage facilities below

the North Train Railway are evaluated as a priority in the drainage basins that are classified as being high risk.

**Author Contributions:** Fatmah Nassir Alqreai and Hamad Ahmed Altuwaijri contributed to the study conception and data collection. Fatmah Nassir Alqreai performed material preparation and analysis. Fatmah Nassir Alqreai wrote the first draft of the manuscript. Also, Hamad Ahmed Altuwaijri added some text to the literature review section and supervised the methodology and results. All authors have read and agreed to the published version of the manuscript.

**Funding:** The authors extend their appreciation to the Deputyship for Research and Innovation, Ministry of Education in Saudi Arabia for funding this research work through the project no. IFKSUOR3-006-2.

**Data Availability Statement:** Not applicable.

**Conflicts of Interest:** The authors declare no conflict of interest.

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
