# Peer review of "Assessing the Hazard Degree of Wadi Malham Basin in Saudi Arabia and Its Impact on North Train Railway Infrastructure"

_ijgi, doi:10.3390/ijgi12090380_

Round 1

Reviewer 1 Report

The paper can be published as this form. 

Author Response

We thank the reviewer for his comment and hope this study inspires more authors in the field.

Reviewer 2 Report

Dear Authors,

please find attached my comments for your paper.

My best regards

Author Response

Review#2 comment 1: Can the bibliographic reference be summarized as mentioned above?

Author response:  We thank the reviewer for the time spent and valuable feedback.

Author action: Yes its already summarized and we use the Journal style.

Review#2 comment 2: Remove the reference. It's mentioned later.

Author response:  We thank the reviewer for the time spent and valuable feedback.

Author action: we cant remove it since it comes between two  different references.

Review#2 comment 3: The main topic of the article should be fully developed in this section, expanding upon the bibliographic references related to the subject.

Author response:  We thank the reviewer for the time spent and valuable feedback.

Author action: The previous studies were expanded from line 98 to 205.

Review#2 comment 4: It is not sequential.

Author response:  We thank the reviewer for the time spent and valuable feedback.

Author action: it is mentioned before in line 107 (semi-quantitative, and quantitative [15]. For example) then reference number 16 comes then we used reference 15 for the next sentence.

(………..the region [16]. As mentioned by [15]…….)

Review#2 comment 5: The authors should provide an analytical definition of the hazard degree.

Author response:  We thank the reviewer for the time spent and valuable feedback.

Author action: We provided this to explain the hazard degree model, this can be seen in line 112 to114 (as an introduction). Then more details in part (5.3)

Review#2 comment 6: Please improve the quality of the image.

Author response:  We thank the reviewer for the time spent and valuable feedback.

Author action: we provided figures with 400 dpi, and the journal instruction mention a minimum 300dpi in (Preparing Figures, Schemes and Tables)

Review#2 comment 7: Uncomprensible.

Author response:  We thank the reviewer for the time spent and valuable feedback.

Author action: it is important to mention all these data and their sources.

Review#2 comment 8: What does X indicate?

Author response:  We thank the reviewer for the time spent and valuable feedback.

Author action: see line 331 (…..X is the morphometric coefficient value used to evaluate each….)

Review#2 comment 9: The flowchart needs to be made clearer. For instance, what are the parameters considered for defining the hazard degree?.

Author response:  We thank the reviewer for the time spent and valuable feedback.

Author action: they mention in the same figure at the top

See this >>>>>Length of overland flow-Weighted mean bifurcation ratio (Rbwm))

Have an inverse relationship with the severity of torrents.

(Basin area-drainage density -Streams frequency Relif ratio-ruggedness number-circularity ratio)

Have a direct relationship with the severity of torrents.

Review#2 comment 10: The authors should specify these parameters

Author response:  We thank the reviewer for the time spent and valuable feedback.

Author action: we added the eight primates as you requested. See line 319

Review#2 comment 11: Please improve the quality of the image.

Author response:  We thank the reviewer for the time spent and valuable feedback.

Author action: we did this based on the journal requirement.

Review#2 comment 12: The authors should better highlight the study's limitations and the potential for developing the results in line with the article's title.

Author response:  We thank the reviewer for the time spent and valuable feedback.

Author action: We mention these restrictions of unit hydrograph this is a limitation of the study and line 563 we focused on basins affecting the Railway in the Wadi Malham area

Reviewer 3 Report

The paper analyzed the degree of risk of 14 drainage basins affecting the North Train railway within the Wadi Malham drainage basin, using the risk degree model with morphometric parameters to determine the degree of hazard in drainage basins. The method used is semi-qualitative using weights and ranks to determine direct surface runoff hazard in drainage basins. The basins are classified into three categories according to estimated risk (low, medium, high). As input data 12.5m DEM and topographic maps were used. The analysis is performed using GIS tools, namely ArcGIS 10.7.

The paper is generally well written with very detail explanations of all theoretical and practical aspects. Research methods and obtained results are clearly presented. Discussion section can be further extended to include implications of the study, limitations, future work and recommendations to the relevant institutions. It is not clear from the conclusion whether better planning is already achieved or it is anticipated in future. How AHP is used? Although average precipitation and temperature are mentioned, it is not clear to what extent weather conditions in the desert climate influence the risk of surface runoffs in the study area and how great the actual risk is.

It should be clarified how the 14 basins were determined? Is it the result of the study or is it the official delineation of river basins (e.g. state mapping agency) of the region? In the former case its accuracy should be assessed.

Somewhere is mentioned that the smallest basin is 2.59km2 and through measurement it is 3.55km2. What caused this difference?

Figures should be placed immediately after the paragraph where they are first mentioned. It is easier for the reader to follow. Furthermore, headings should not be placed at the end of the page (line 338) or figure captions on the next page (line 376).

Referencing of tables are in some cases incorrect (e.g Table 3, instead of Table 2 – line 450, line 461, line 477, line 487, line 500). It is advisable to check all the referencing of figures and tables.

Line 535 This study used spatial data and technologies as much as possible. – the sentence should be rephrased   

There are some typos that need to be corrected, e.g. line 25 . .

Line 221 replace astronomically with geographically

Line 450 Figure 10 , that

Figure 11. ruggedness – capital letter

range between 0.44 477 and 0.27 – vice versa, from min to max

range from 0.0899 km and 0.0795 km - vice versa, from min to max

Line 515 – left side of equation is missing

Line 576 basins; Which

Check for other typos as well

English is fine. Minor editing is required. Formatting and typos should be corrected.

Author Response

Review#3 comment 1: It is not clear from the conclusion whether better planning is already achieved or it is anticipated in the future. How AHP is used? Although average precipitation and temperature are mentioned, it is not clear to what extent weather conditions in the desert climate influence the risk of surface runoffs in the study area and how great the actual risk is.

Author response:  We thank the reviewer for the time spent and valuable feedback.

Author action: This study participated in coming up with a priority area that should be focused on then it is recommended in line 576 that more study should be done to have better planning to avoid the risk ……

  • The AHP is a different method that only covers it in literature.
  • This study focused on the morphometric parameters.

Review#3 comment 2:  It should be clarified how the 14 basins were determined? Is it the result of the study or is it the official delineation of river basins (e.g. state mapping agency) of the region? In the former case its accuracy should be assessed.

Author response:  We thank the reviewer for the time spent and valuable feedback.

Author action: They were identified based on the study and verified for accuracy as its used in topographical maps studies.

Review#3 comment 3: Somewhere is mentioned that the smallest basin is 2.59km2 and through measurement it is 3.55km2. What caused this difference?

Author response:  We thank the reviewer for the time spent and valuable feedback.

Author action: Yes 2.59 is the smallest basin that can be studied due to the restrictions of unit hydrograph theory which mention in references number (39) Bedient, F.; Huber, W.; Vieux, B. Hydrology and Floodplain Analysis; King Saud University Publishing House, 2020.

But in the study area, we found the smallest basin was 3.55 which is above the 2.59 (unit hydrograph theory)

Review#3 comment 4: Figures should be placed immediately after the paragraph where they are first mentioned. It is easier for the reader to follow. Furthermore, headings should not be placed at the end of the page (line 338) or figure captions on the next page (line 376).

Author response:  We thank the reviewer for the time spent and valuable feedback.

Author action: We fixed based on Reviewer's comments, but sometimes the paragraph need to be complited then we insert the figure such as fig1.

Review#3 comment 5: Referencing of tables are in some cases incorrect (e.g Table 3, instead of Table 2 – line 450, line 461, line 477, line 487, line 500). It is advisable to check all the referencing of figures and tables.

Author response:  We thank the reviewer for the time spent and valuable feedback.

Author action: thanks we fixed it based on your comments

Review#3 comment 6: Line 535 This study used spatial data and technologies as much as possible. – the sentence should be rephrased  

Author response:  We thank the reviewer for the time spent and valuable feedback.

Author action: We rephrased.  

Review#3 comment 7: There are some typos that need to be corrected, e.g. line 25 . .

Line 221 replace astronomically with geographically

Line 450 Figure 10 , that

Figure 11. ruggedness – capital letter

range between 0.44 477 and 0.27 – vice versa, from min to max

range from 0.0899 km and 0.0795 km - vice versa, from min to max

Line 515 – left side of equation is missing

Line 576 basins; Which

Check for other typos as well

Author response:  We thank the reviewer for the time spent and valuable feedback.

Author action: thanks, we fixed based on your comments

Round 2

Reviewer 2 Report

Dear Authors,

I thank you for your care and attention to the revisions. I believe that some corrections and additions have been made in line with the suggestions I provided. Therefore, the article can be accepted for publication.

Best regards

Author Response

Thanks for your time. 

Reviewer 3 Report

The authors responded appropriately to the previous round of review.

Author Response

Thanks for your time.